## REVIEW ARTICLE

# Towards organoid culture without Matrigel

Mark T. Kozlowski [1✉], Christiana J. Crook [2,3,4] & Hsun Teresa Ku [2,3]

Organoids—cellular aggregates derived from stem or progenitor cells that recapitulate organ function in miniature—are of growing interest in developmental biology and medicine. Organoids have been developed for organs and tissues such as the liver, gut, brain, and pancreas; they are used as organ surrogates to study a wide range of questions in basic and developmental biology, genetic disorders, and therapies. However, many organoids reported to date have been cultured in Matrigel, which is prepared from the secretion of Engelbreth-Holm-Swarm mouse sarcoma cells; Matrigel is complex and poorly defined. This complexity makes it difficult to elucidate Matrigel-specific factors governing organoid development. In this review, we discuss promising Matrigel-free methods for the generation and maintenance of organoids that use decellularized extracellular matrix (ECM), synthetic hydrogels, or gel-forming recombinant proteins.

Organoids are multicellular structures derived from stem and progenitor cells that mimic the function and spatial organization of organs[1]. Organoids recapitulate important organ functions in vitro while remaining small in size and often free of interfering cell types such as vascular, nerve, or other undesired epithelial cells. For these reasons, organoids are used to study organ development[2] and model various diseases such as cancers[3], neural disorders[4], and autism[5]; they are also used as pharmaceutical testing platforms[6], model systems for CRISPR-CAS9-mediated treatment of genetic diseases[7], and replacement organs for transplantation[8,9]. It is possible to culture organoids from induced pluripotent stem cells (iPSCs)[10] or adult stem cells from patient tissues[11], which may lead to personalized medicine. The wide range of clinical applications of organoids is the subject of a recent review by Drost and Clevers[12]. Many excellent reviews have been published about different organoid types, such as heart[13,14], brain[15–17], liver[18,19], kidney[20–22], pancreas[23–25], and female reproductive tract[26].

Many organoids have been cultured in Matrigel, a material derived from the secretion of Engelbreth–Holm–Swarm mouse sarcoma cells and enriched for extracellular matrix (ECM) proteins[27]. In an early report of organoid culture, Sato and colleagues grew murine intestinal Lgr5[+] stem cells in high concentrations of Matrigel supplemented with the growth factors WNT, Noggin, R-spondin, and EGF[28]. This culture system has been widely adapted for other organs such as the colon, stomach, and liver[29–33]. Related methods have been used to construct simulated versions of the inner ear[34] and pancreas[35–39], and the similarities between pancreatic and liver-organoid-generating cells suggest that methods used for making liver organoids may be applicable to the pancreas[40]. The many organoid-specific applications of Matrigel-based culture methods have been thoroughly discussed elsewhere[41–43].

Despite its versatility and affordability, Matrigel is extremely complex; proteomic analysis shows that it contains more than 1800 unique proteins[44]. The undefined nature of Matrigel makes it difficult to identify the signals necessary for organoid structure and function; this difficulty is compounded by lot-to-lot variations of Matrigel[45–48]. Furthermore, Matrigel may

[1]DEVCOM US Army Research Laboratory, Weapons and Materials Research Directorate, Science of Extreme Materials Division, Polymers Branch, 6300 Rodman Rd. Building 4600, Aberdeen Proving Ground, Aberdeen, MD 21005, USA. [2]Department of Translational Research and Cellular Therapeutics, Diabetes and Metabolism Research Institute, City of Hope National Medical Center, 1500 Duarte Rd., Duarte, CA 91010, USA. [3]Irell and Manella Graduate School of Biological Sciences, Beckman Research Institute of City of Hope, 1500 Duarte Rd., Duarte, CA 91010, USA. [4]Present address: Department of Medical Oncology and Therapeutics Research, City of Hope National Medical Center, 1500 Duarte Rd., Duarte, CA 91010, USA. ✉email: mark.t.kozlowski4.civ@army.mil

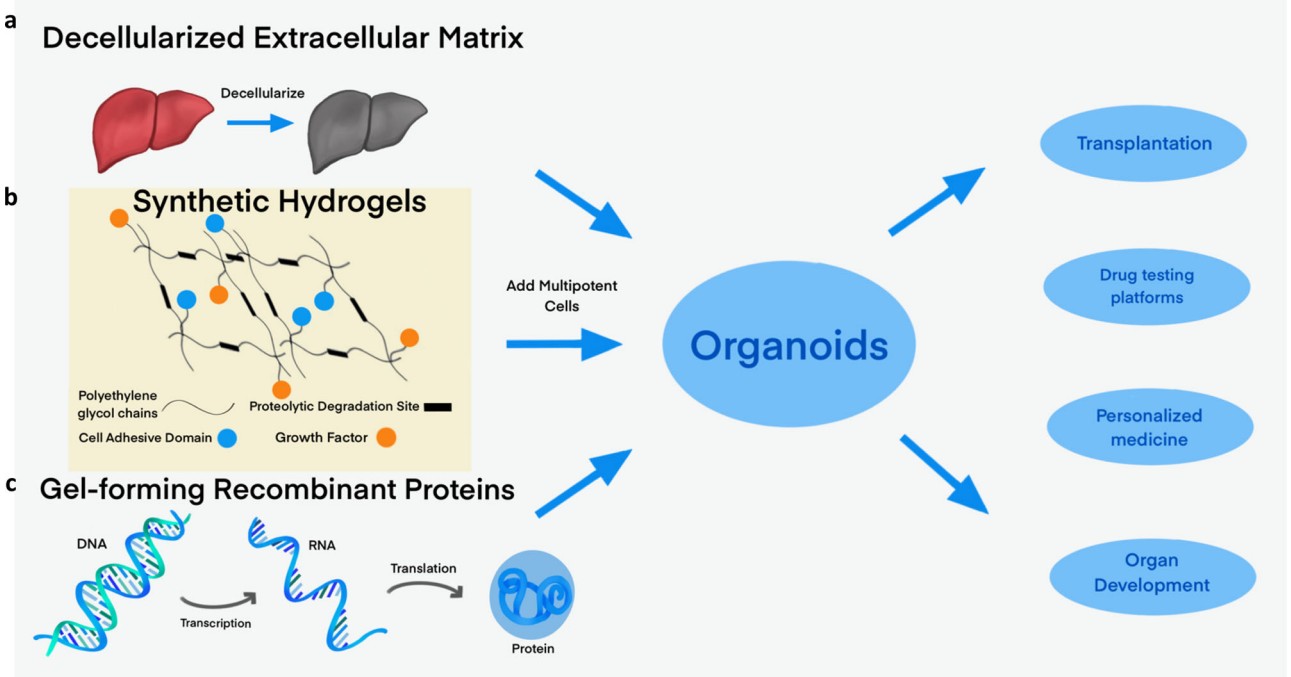

**Fig. 1 Methods of making organoids without Matrigel.** Replacing the undefined medium of Matrigel is a major goal of organoid culture. We will discuss three main alternative media: (**a**) decellularized extracellular matrix and other derived proteins, (**b**) synthetic hydrogels, which generally incorporate cell-adhesive domains or proteolytic degradation sites, and (**c**) gel-forming recombinant peptides. Adding multipotent cells to these matrices enables the growth of organoids, which are potentially applicable as transplants, drug-testing platforms, personalized medicine, and means to understand organ development.

not contain all of the necessary components for proper organoid formation; gut organoids cultured in Matrigel lack the characteristic architecture of mammalian intestines, which could be due to a sub-optimal amount of laminin-511 and the absence of other cell types such as mesenchymal cells[49,50]. Finally, it has become increasingly clear that the mechanical properties in three-dimensional (3D) culture systems can have large effects on cell[51], organoid[52], tissue[53], and organ development[54,55]. The mechanical properties, such as elastic modulus, pore size, stress relaxation, and creep[56–59], cannot be easily separated from the chemical cues in the Matrigel-based culture systems. Furthermore, the mechanical properties of Matrigel samples are heterogeneous; local regions of such gels have been found to exhibit elastic moduli several times higher than the average modulus of the sample[60,61]. Finally, the fact that Matrigel is originated from mouse cells hampers its use in human clinical transplantation due to potential immunogenicity[18].

Given these limitations, there is an emerging need to develop Matrigel-independent organoid culture methods. In this review, we discuss recently-developed Matrigel-free techniques for the culture of organoids. We will review undefined matrices, focusing on ECM derived from decellularized tissues and collagen, and defined matrices, including synthetic polymer hydrogels and engineered ECM proteins (Fig. 1). Table 1 summarizes the advantages and disadvantages of each category of material. Table 2 summarizes studies that discuss the effects of elastic modulus on the organoid formation of various tissues.

## Organoid culture in decellularized ECM and other naturally-derived proteins

In organ development, ECM proteins provide signaling cues, serve as an adhesive substrate, and sequester growth factors (Fig. 2)[62]. In order to accurately recapitulate the composition,

structure, and vascularization of native ECM in organ development, some organoids have been grown in decellularized ECM from human or animal donors. The methods of decellularization used are dependent on the target tissue and not readily generalizable; a number of these methods have been reviewed elsewhere[63]. While xenogeneic ECM has the potential to cause immune responses, this risk can be greatly reduced by using proper preparation techniques[64]; similar ECM scaffolds derived from animals are FDA-approved for clinical applications such as heart valve replacement, facial reconstruction, and osteopathic implants[62,65]. Decellularized ECM may also provide additional cues that promote regeneration of damaged tissue, ultimately supporting the organoid transplant and promoting its function[66]. Decellularization approaches have been demonstrated for human kidney[67], murine kidney[67,68], murine heart[69,70], human and porcine lung[71], and porcine testicular[72] tissues, with each type posing unique challenges. To illustrate some of these challenges and methods, we will focus on the decellularization of the liver, gut, and pancreas.

**Liver organoids grown in decellularized ECM.** Liver-specific ECM can be obtained from a surgically resected portion of a patient's damaged liver, or from livers unsuitable for transplantation. Lin and colleagues reported that liver tissue decellularization supported growth and maintenance of rat hepatocytes; however, this method relied on mechanical disruption of resected tissue, which resulted in the loss of organ architecture and vascular networks[73]. In contrast, Baptista and colleagues perfused Triton X-100 and ammonium hydroxide through a ferret hepatic vascular network to remove cells. This method preserved the underlying ECM and vasculature while retaining most of the glycosaminoglycans, collagens, and elastins. The decellularized material could be colonized by human fetal liver and endothelial

**Table 1 Different types of materials for the generation of organoids from various tissues in three-dimensional culture.**

| Materials | Advantages | Disadvantages | Organoids made using this type of material | References |
|---|---|---|---|---|
| Matrigel | Inexpensive and commercially available, extensively used with well-developed protocols | Undefined culture system, subject to lot-to-lot variation, poor control of mechanical properties, may not contain all chemical cues necessary for differentiation, immunogenicity | Gut, heart, brain, liver, kidney, pancreas, female reproductive tract, inter alia. | See references[12-27] for reviews |
| Decellularized tissue | Preserves native chemical cues and mechanical properties, resulting organoids can be large | Preparation is difficult, limited by donor availability, lack of definition | Liver, intestine, heart, lung, kidney, pancreas, testicular, stomach | Liver:[73-77,80] Intestine:[79,80] Heart:[69,70] Lung:[70,71,121] Kidney:[67,68,70] Pancreas:[80-82] Testes:[72] Stomach80: |
| Collagen and other biomacromolecules derived from natural sources | Low cost, wide availability | No structural information preserved, not all necessary chemical cues present, often requires feeder cells, lot-to-lot variation | Liver, intestine, pancreas, epithelium, brain, lung, vascular, stomach, kidney | Liver:[98] Intestine:[84-86,89-91,102,103] Pancreas:[104] Epithelium:[86,88,89] Brain:[101,111-116] Lung:[99,100] Vascular:[105] Stomach:[85,102] Kidney:[87] |
| Synthetic polymers | Excellent control of mechanical and chemical properties, repeatability, tunable degradation rate | Requires functionalization with cell-binding peptides or presence of feeder cells, possible cytotoxicity concerns | Brain, liver, intestine, pancreas, salivary glands | Neural and Brain:[156,157,161,165,169,174] Liver:[158-160,179] Intestine:[162-164] Pancreas:[175] Salivary Glands:[183] |
| Recombinant proteins and peptides | Precise placement of chemical cues, tunable mechanical properties, tunable degradation rate, easy to include cell-binding domains | Possible endotoxin contamination, higher cost, possible immunogenicity | Pancreas, brain, intestine, heart | Pancreas:[192-195] Brain:[196,198-202] Intestine:[191] Heart:[189,190] |

cells to produce a functioning organoid[74]. An illustration of the method of Baptista and coworkers is shown in Fig. 3. The liver can also be decellularized, ground into powder, and redissolved. Lee and colleagues used this approach with rat liver ECM to promote the differentiation of human adipose-derived stem cells into functional hepatocytes[75]. More recently, Saheli and colleagues seeded sheep liver ECM gel with a combination of human hepatocarcinoma cells, mesenchymal stem cells, and umbilical cord stem cells; the resulting tumor organoids had greater hepatocyte function than tumor organoids grown in comparable collagen I-based culture[76]. Lewis and colleagues observed that growing murine small cholangiocytes (a committed progenitor cell type) in porcine liver ECM gel resulted in the formation of complex, branching structures similar to biliary ducts, and these cells also secreted small amounts of bile. In contrast, cholangiocytes cultured in Matrigel formed cysts while those in collagen I proliferated and spread in all directions without spontaneously forming structures[77]. Thus, Matrigel alone does not provide all of the needed factors for small cholangiocyte differentiation,

whereas a decellularized liver ECM gel appears to be a better alternative.

**Gut organoids grown in decellularized ECM.** Decellularized matrices have also been used for the growth of human intestinal organoids (HIOs) derived from pluripotent stem cells (PSCs), as well as enteroids derived from adult crypt stem cells[78]. Finkbeiner and colleagues found that undifferentiated human embryonic stem cells (ESCs) could not directly differentiate into HIOs in a decellularized porcine intestinal ECM. However, pre-differentiated HIOs were able to seed onto decellularized porcine intestinal ECM and form correct spatial orientation-mimicking intestine[79]. Giobbe and colleagues developed a method for decellularizing the porcine small intestine to form an intestinal ECM gel similar to the liver ECM gels discussed in the previous section. The porcine intestinal ECM gel was able to support the formation of enteroids from murine Lgr5+ crypt cells, and from human pediatric stomach and intestinal crypts. The authors were also able to achieve mechanical control of the ECM gel by incorporating poly-acrylamide to achieve different stiffness for two-dimensional (2D) culture of human and mouse enteroids, which may be important for future research. This intestinal ECM gel is also applicable to grow organoids from the liver, stomach, and pancreas[80].

**Decellularized ECM from the pancreas.** Decellularized ECM has been prepared from various pancreatic cell sources, such as adult human pancreas[81] and porcine pancreas[82]. Using mass spectrometry, Bi and colleagues found major differences in protein compositions comparing decellularized rat pancreatic extracellular matrix to Matrigel; Matrigel contains lower levels of collagen V than are normally present in pancreatic ECM. By coating plates with Matrigel plus commercially available collagen V, the authors were able to enhance endocrine differentiation of human iPSCs in 2D culture, compared to Matrigel alone[83]. One key limitation of this study, however, is that the proteomic profile

**Table 2 Ideal elastic moduli for generating organoids from different organs.**

| Organoid | Ideal matrix elastic modulus | References |
|---|---|---|
| Mouse ESC-derived neuroepithelial | 2–4 kPa | 156,157 |
| Mouse liver | 6–20 kPa | 179 |
| Human intestine | 100–200 Pa | 162-164, |
| Mouse intestine | 100–200 Pa | 191 |
| Human brain | 100 Pa-1 kPa | 51,114,197 |
| Human fore-brain | 300 Pa | 116 |
| Human hindbrain | 1 kPa | 116 |
| Mouse heart | 700 Pa | 190 |
| Mouse bone | 34 kPa | 51 |

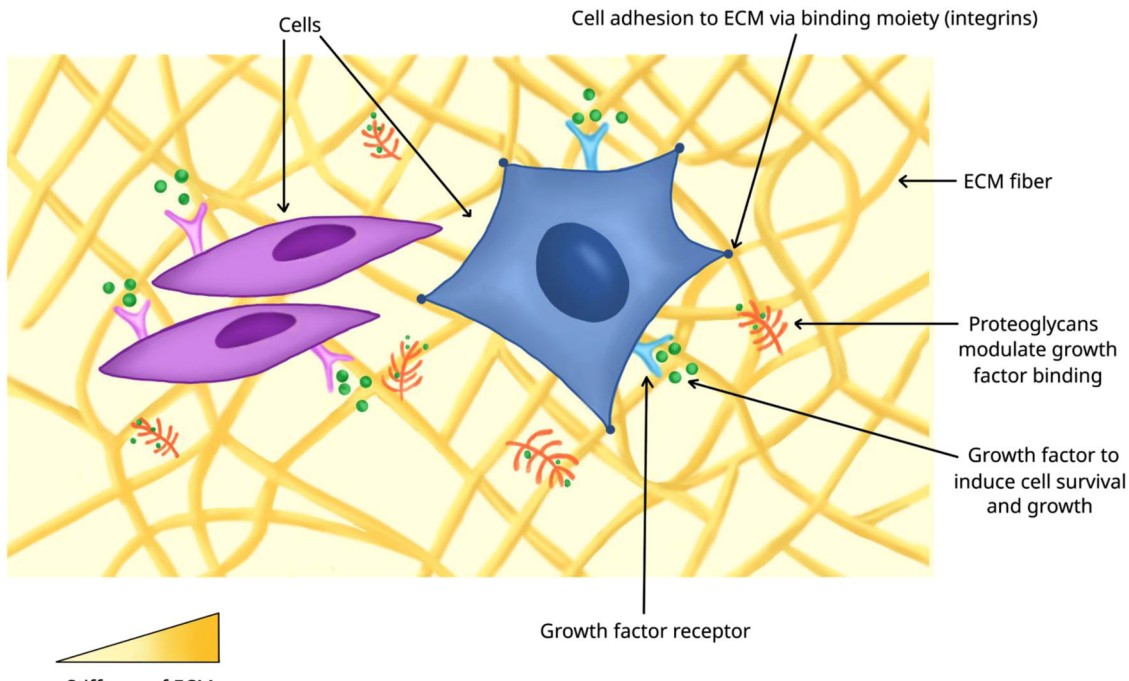

**Fig. 2 Microenvironment of cells.** Cells in an organ or organoid are surrounded by other cells, extracellular matrix (ECM) proteins, and growth factors sequestered in the proteoglycan-modified ECM proteins. Cells bind to ECM proteins via adhesion molecules, such as integrin receptors, which provide signaling cues to exert biological functions. The stiffness of ECM experienced by the cells also affects their biological activities.

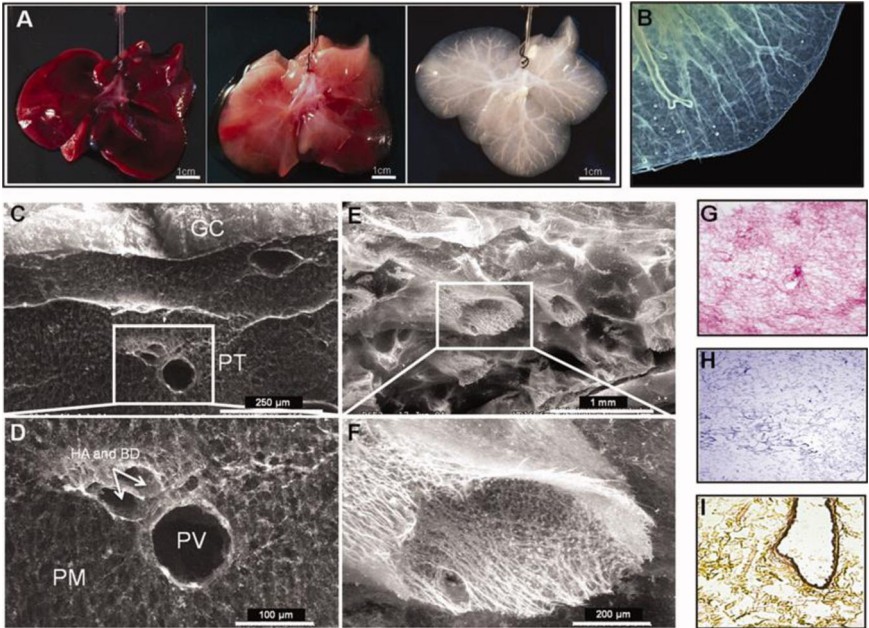

**Fig. 3 An example of whole-organ ferret liver decellularization with excellent retention of structural information, for use as an organoid scaffold.**
Figure (**a**) shows the liver at the start of treatment, then after 20 and 120 min of decellularization. A micrograph of the decellularized liver, (**b**), shows that the liver capsule and vasculature remain intact after cell removal. Scanning electron microscope images show that the structure of the liver is remarkably well-conserved, with an intact Glisson's capsule (GC) visible in (**c**), and an intact hepatic artery (HA), hepatic portal vein (PV), and biliary duct (BD) visible in (**d**). Other blood vessels are structurally intact, despite cell removal (**e**), with the structural details apparent even at high magnification (**f**). H&E staining (**g**) shows that all cells have been removed, with the pink stain showing protein-containing extracellular matrix; this absence of cellular material is further confirmed by Mason's Trichrome staining (**h**). Movat-Pentachrome staining (**i**) shows the presence of collagen in yellow, and a dark stain shows elastin around an artery. Decellularization can proceed gently enough to retain structural information, yielding scaffolds that can be colonized by pluripotent cells which then differentiate into mature organoids. The image is from ref. [74] and is reproduced with permission.

of the rat pancreas may be very different from that of the human pancreas. The human pancreas presents a challenge in decellularization as it has a higher lipid content than animal models. With specific preparation methods that remove lipids, Sackett et al. found that decellularized human ECM is capable of supporting the survival of undifferentiated hPSCs and their pancreatic lineage derivatives, including insulin-expressing beta-like cells, in vitro[81].

**Tissue organoids grown in other naturally-derived proteins and biomacromolecules**. A complementary approach to the use of decellularized ECM is naturally-derived proteins, such as collagen I derived from porcine tendon, porcine skin, or bovine lens capsules. Collagen I has been used to form human colorectal carcinoma model organoids in a 3D culture of rabbit colons[84] and as a support for the culture of human and murine intestinal, stomach, and colonic organoids[85]. Other examples are vitrified collagen I for human intestinal organoids[86] and murine renal organoids[87], as well as fibrin supplemented with laminin-111 capable of supporting various murine and human epithelial organoids[88].

The Tokyo Medical and Dental University (TMDU) method uses collagen I for intestinal enteroid culture[89]. Yui and colleagues showed that embedding intact murine colonic crypts and isolated Lgr5+ progenitor cells in collagen I with hepatocyte growth factor, R-spondin 1, EGF, and Noggin generated organoids that were able to engraft onto damaged mouse intestinal epithelia upon transplantation. In contrast to the TMDU method, the Ootani method uses collagen I gel in which small and large intestinal cells are kept suspended at an air-liquid interface; this method improves oxygenation of the organoid,

allows viable murine organoids to be maintained in culture for up to 350 days, and preserves the mesenchymal niche in the organoids[90].

Two factors have been shown to affect the differentiation of progenitor cells into organoids in collagen-based matrices: the source of seeded cells, and the spatial arrangement of collagen types around the cells. Isshiki and colleagues reported that the choice between the TMDU and Ootani methods should be governed by the source of cells used to generate the intestinal organoid. More consistent results are obtained for growing intestinal organoids from seeded isolated rat intestinal crypts using the TMDU method, whereas the Ootani method is better for growing rat colon organoid cultures of homogenized tissue[91].

In addition to judicious selection of cell type, cell fate is intimately tied to interactions between cell surface integrins and biochemical cues in the ECM[92]. Collagen I has a high affinity for $\alpha_2\beta_1$ integrin, whereas collagen IV binds more strongly to $\alpha_1\beta_1$ integrin[93]. Collagen IV tends to occur exclusively in basement membranes[93]. $\beta_4$ integrin expressed by intestinal organoids is distributed only at the basal surface[94], while $\beta_1$ integrin is required for proper apical-basal polarization[95–97]. A combination of collagen I and fibronectin compared to collagen I alone functionalized within a PEG gel enhanced hepatic differentiation from human mesenchymal stem cells[98]. Clearly, the culture of organoids must take the 3D spatial positioning of the relevant materials into account.

In addition to organoids grown in naturally-derived proteins, several laboratories have grown a wide range of organoids in polysaccharides such as alginate or alginate-chitosan mixtures. Organoid types grown in alginate include human lung[99,100], human brain[101], murine intestinal[102], human intestinal[102,103], human pancreatic[104], and human and murine vascular[105].

Capeling and colleagues grew HIOs on an alginate substrate and found that differentiation of human pluripotent cells into HIOs could be supported without the alginate providing chemical cues to the cells. The authors hypothesized that cells create their own niche within the alginate hydrogel by secreting basement membrane proteins and forming mesenchyme, allowing cellular survival and differentiation into HIOs[103]. Rossen and colleagues demonstrated the development of murine and human vascular organoids in a non-functionalized alginate setting[105]. Alginate has a number of advantages that make it attractive as a material for further study; it is inexpensive, relatively easy to modify and functionalize, biocompatible, and has been used in a wide range of biological and materials applications[106–108]. The mechanical properties of alginate, such as elastic modulus, extensibility, and characteristic relaxation time, can also be easily tuned[109]. For these reasons, alginate is a promising material for further exploration. However, because alginate is biologically derived, its mechanical properties are still subject to lot-to-lot variability[110]. Similarly, hyaluronic acid and mixtures of hyaluronic acid and chitosan have been extensively used in the growth and construction of neural organoids[111–116].

**Advantages and disadvantages of decellularized ECM and other naturally-derived proteins or biomacromolecules.** Decellularized ECM-based methods can quickly recapitulate organ function. Many or all of the chemical cues required for the formation of a spatially-defined organ, including difficult-to-introduce glycoproteins, are already present, minimizing the need for additional chemical modification of the ECM. Decellularized ECM retains the compositional differences observed between basal and apical regions. Collagen- and alginate-based materials have been approved by the FDA for a wide variety of applications[117], which allows for rapid clinical translation.

Decellularized ECM does have disadvantages. Most importantly, the quantity of ECM that is available for study is limited by the availability of donor animals or humans, and the quality of ECM can be affected by the health of a donor. For example, emphysematous or fibrotic lung tissue has hardened and undergone alterations in its architecture. These alterations can lead to cells failing to survive beyond one week of culture[118] or broad changes in the phenotype of seeded cells that do survive[119]. Contrarily, myocardial infarct is known to trigger remodeling events that stiffen the ECM and change its chemical composition; yet when mesenchymal stem cells are seeded on infarcted tissue, the cells secrete higher levels of pro-survival and immunomodulatory growth factors[120]. While myocardial infarct appears to enhance the survival of seeded cells, the negative effects of other diseased tissue on organoid development should not be discounted.

Even with healthy donor tissue, batch-to-batch variability remains. The physical properties of decellularized ECM are difficult to control or modify, which limits the experiments that can be conducted. Decellularized ECM is also chemically undefined; the factors driving differentiation are often unknown. Surface proteoglycans that are necessary for successful organoid formation may be removed by harsh decellularization[121]. A related difficulty is that not all decellularization protocols are equally effective at removing cells or other immunogenic species, which can cause varying host immune responses and failure of implants in clinical trials[122]. Finally, the occasional need for PSC differentiation into organ-specific progenitor cells that are then introduced into the decellularized matrix requires an additional step.

Collagen-based culture methods are not limited by donor tissue availability; biomedical-grade collagen can be harvested on an industrial scale from cows and pigs. However, some collagen-based culture methods rely on coculture with supporting cells[98], which introduces undefined components into the organoid culture. Furthermore, it is difficult to modify the mechanical properties of these culture systems without altering chemical concentrations. To elucidate the effects of mechanical properties on organoid development, researchers have turned to synthetic hydrogels that have been functionalized with cell-binding cues.

## Organoid culture in synthetic hydrogels

Native ECM is complex; it contains over 300 different proteins, each of which has a different biological function and stiffness[123]. This large number of proteins means that many variables cannot be easily dissected to study the influences of ECM on organoid behavior and development. Synthetic hydrogels are attractive because their mechanical properties, functionality, and erosion rate can be controlled. The matrix metalloprotease (MMP) family of enzymes affects cellular and organoid development by degrading ECM proteins[124]. By including MMP recognition sites on synthetic hydrogels, it is possible to tune the rate of the hydrogel's erosion. Manipulating synthetic hydrogels using methods such as electrospinning[125], photopatterning[126,127], spraying of microspheres[128], inkjet and 3D printing[129], or microfluidic channels[130] further enables control over the shape and size of the organoids. The ability to exert local control over chemical and mechanical properties allows researchers to duplicate the heterogeneity in stiffness and composition found in organs, generate interfaces between materials similar to those found in the ECM, and duplicate essential elements of material microstructure; each of these controls has implications for organ function and disease[131,132]. Synthetic hydrogels can also be made responsive to external stimuli. For example, a thermoreversible hyaluronic acid- poly($N$-isopropylacrylamide) (PNIPAAm) based hydrogel that solidifies at 37 °C and re-liquefies upon cooling enabled culture and recovery of human pluripotent stem cells without enzymatically digesting the matrix[133]. Light-sensitive polyvinyl alcohol matrices have recently been developed for cell culture; these matrices allow for control over the spacing of biochemical cues and the material environment[134]. Both of these materials may be useful for future organoid studies.

The use of synthetic hydrogels may also open up new avenues by altering the porosity of the scaffold on which the cells are grown. Dye and coworkers found that human lung organoids transplanted in mice could merge together to form airway structures. However, this was only possible if the scaffold was able to degrade, which increased the material's pore size[135]. Choi and coworkers reviewed the pore sizes typically used in making tissue-engineering scaffolds, ranging from 5–15 microns for fibroblasts to 200–400 microns for osteoblasts, and developed a method for creating an artificial kidney scaffold using microstereolithography[136]. However, these studies observed the effects of pore size on mature cells rather than organoid development from multipotent cells. Broguiere and coworkers found that a Matrigel culture system had a pore size smaller than 200 nm, or the resolution limit of their confocal microscope, but that a fibrin-based material had a pore size closer to 4 microns; these materials had comparable elastic moduli and colony-forming efficiency[88]. To our knowledge, studies showing an explicit connection between pore size and organoid differentiation have not yet been reported.

Synthetic hydrogels also have readily tunable viscoelastic properties, such as loss modulus and characteristic relaxation time. Many relevant tissues are viscoelastic, with brain tissue in particular not only having a strong dissipative component (i.e., high loss modulus) in its response to stress[137] but also having slight differences in viscoelastic properties between white matter and gray matter[138]. The viscoelastic properties of a material affect

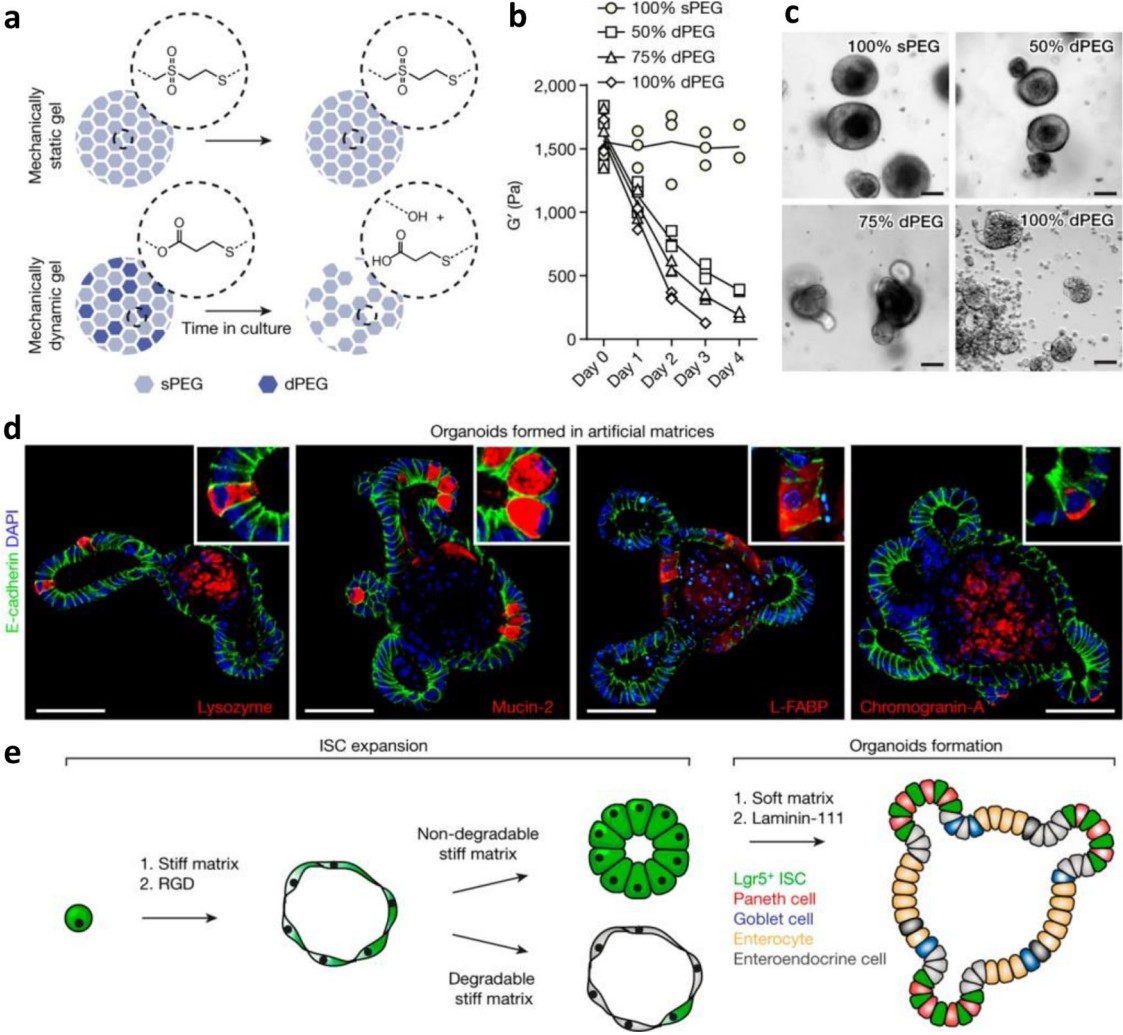

**Fig. 4 Growth of intestinal organoids on synthetic hydrogels, and effects of matrix stiffness and degradability on their formation.** Gjorevski and colleagues demonstrated defined PEG-based intestinal organoid culture. The stiffness and degradability has a major effect on the ability of induced pluripotent cells to differentiate into intestinal organoids. By varying the ratio of hydrolytically labile functional groups (dPEG) to stable functional groups (sPEG), the rate of degradation of the gel can be controlled (**a**, **b**). Higher ratios of sPEG are associated with the expansion of intestinal stem cells, whereas the degradable gels lead to the formation of organoids containing differentiated cells (**c**). In fact, organoid formation is observed only in gels that have a stiffness of ~190 Pa: cells expressing lysozyme (Paneth cells), mucin-2 (Goblet cells), and Chromogranin-A (enteroendocrine cells) are in different compartments, indicating that specialized cells are spatially separated (**d**). In short, a stiff matrix leads to intestinal stem cell proliferation and expansion, but a soft matrix and functionalization with laminin-111 promotes differentiation (**e**). The image is from ref. [162] and is reproduced with permission.

matrix remodeling, cell spreading, migration, differentiation, and consequently, organoid fate[139–141]. The effects of viscoelastic properties on cell culture and behavior are complex but have been thoroughly reviewed elsewhere[140,142]. The effects of materials properties other than stiffness have only recently begun to be explored, and tunable hydrogels will enable more sophisticated experiments to be conducted.

**The role of chemical cues in organoid differentiation.** Synthetic polymer-based culture allows organoid formation conditions to be evaluated using high-throughput methods[143–145]. Synthetic hydrogels can be functionalized with biologically active moieties that permit the growth and spread of cells[146–151]; concentration and spacing of these cues can be changed independently[152–155]. A striking demonstration of the utility of high-throughput approaches was provided by Ranga and colleagues, who prepared PEG-based gels in 1536-well plates and studied murine ESCs (mESCs) expressing an Oct4-GFP reporter. The authors analyzed 1000 variations of matrix elastic modulus, cell-binding

peptides, and matrix susceptibility to MMP degradation and their effects on murine ESC fate[156]. The optimal conditions (elastic moduli ranged from 2–4 kPa and scaffolds with MMP insensitivity) produce murine neural tube organoids that are more homogenous in colony size and morphology, as well as more polarized, than those grown in Matrigel. The percentage of cells containing an actomyosin contractile ring is used as a metric for the polarity of the cells[157].

Cell-binding cues from collagen, fibronectin, or laminin have frequently been added to synthetic hydrogels to allow for organoid growth and differentiation. Ng and colleagues created functional human liver organoids derived from human iPSCs in a colloidal crystal of PEG functionalized with collagen I, fibronectin, or laminin-521[158,159]. Attachment of human iPSCs was successful in assemblies functionalized with collagen I and laminin-521 but not with fibronectin. This result builds on previous work by the authors, in which basic human liver function was recapitulated by a PEG-based scaffold[160]. To promote iPSC differentiation into human neuronal progenitor

cells, Ovadia and colleagues compared photo-crosslinked PEG-based gels that contained chemical cues such as the laminin-derived cell-binding sequences YIGSR and IKVAV, the fibronectin-derived sequences PHSRNG$_{10}$RGDS and RGDS, and the vitronectin-derived sequence KKQRFRHRNRKG[161]. The authors found that PEG gels functionalized with YIGSR and PHSRNG$_{10}$RGDS were permissive for human iPSC survival and differentiation into neural progenitor cells in 3D culture.

**The role of stiffness in organoid differentiation**. The stiffness of synthetic hydrogels can be controlled and has an effect on organoid formation. Gjorevski and colleagues reported a synthetic matrix for the intestinal organoid culture of murine and human Lgr5$^+$ progenitor cells derived from the intestinal crypt. The material consisted of a PEG gel functionalized with either an RGD fibronectin-derived peptide or a laminin-111-derived peptide[162]. A stiffer matrix containing an RGD fibronectin-derived peptide promoted survival and proliferation of undifferentiated progenitor cells. In contrast, the softer matrix containing a laminin-111-derived peptide promoted the differentiation of progenitor cells into functional murine and human organoids. The authors found that organoid formation in minimal nutrient conditions was permissible only within a narrow range of matrix stiffness; the optimal elastic modulus was 190 Pa. Making the PEG gels more susceptible to MMP degradation resulted in depolarized organoids with irregular shapes. Major findings of this study are illustrated in Fig. 4, which also demonstrates the effect of matrix stiffness and degradability on the formation of one class of organoids.

Cruz-Acuna and colleagues also found that PEG-based materials may need to be soft and degradable in order to support differentiated intestinal organoids. The authors used a 4-arm PEG maleimide to encapsulate and culture Matrigel-derived HIOs. Organoid viability was reduced at high PEG density and matrix stiffness; RGD- or AG73 (CGGRKRLQVQLSIRT)-functionalized PEG matrices promoted greater organoid viability than laminin-derived IKVAV- or type I collagen-derived GFOGER peptides[163,164]. After successful engraftment, HIOs generated from 4-arm PEG maleimide promoted the healing of mucosal wounds in a mouse colon injury model. Intriguingly, when PEG was crosslinked with dithiothreitol to inhibit matrix degradation, organoid viability at seven days was poor as measured by live-dead staining. This demonstrates a requirement of a degradable matrix for prolonged survival.

**Synthetic hydrogels in action: Modeling difficult tissues such as the brain and pancreas**. Neural organoids present special challenges including a lack of reproducibility, batch-to-batch variation in transcriptional profiles, and susceptibility to small microenvironmental changes that may have a considerable effect on organoid fate[165].

Essential elements of the complexity of the brain must be recapitulated to enable clinical applications of neural organoids. For example, toxicity in brain tissue has many potential causes involving multiple cell types[166–168]. Therefore, the most useful organoids for toxicity models should include multiple populations. Schwartz and colleagues used a PEG-based gel functionalized with pendant RGD cell-binding domains and crosslinked by MMP-degradable peptides to generate neural organoids[169]. In this study, cells were introduced in three sequential stages: neural cells were introduced at day 0, vascular and mesenchymal stem cells at day 9, and microglia and macrophage precursors at day 13. The organoids were then exposed to a library of known toxic and nontoxic compounds, and the resulting RNA-seq data of the organoid response were used to build a machine-learning algorithm to assess the neurotoxicity of known and unknown compounds. In a blinded test, nine out of ten tested

chemicals were correctly identified as toxic or nontoxic; in contrast, the true positive rate of chemical identification in animal models is between 41 and 71%[170].

Brain organoids cultured in hyaluronic acid hydrogels were used to model Down syndrome by Wu and colleagues[114]. The authors found that the differentiation of both normal and Down syndrome iPSCs into neurons was dependent on matrix stiffness; cells could be grown at a softer elastic modulus of ca. 500 Pa but not ca. 1500 Pa, as indicated by higher expression of β-2 tubulin and microtubule-associated protein 2. However, Down syndrome patient-derived iPSCs that were differentiated in the softer gel showed no discernable neurite outgrowth, suggesting a block in the maturation of the differentiated neurons.

Hyaluronic acid hydrogels can also be functionalized with various peptides to examine brain organoid differentiation. Lam and colleagues found that the concentration of laminin-derived IKVAV, with 300 μM being the optimal concentration, was critical to neural organoid survival; however, this concentration did not enhance neuronal differentiation[115]. Bejoy and colleagues showed that functionalizing hyaluronic acid with heparin affects neuronal patterning; the addition of heparin favored differentiation of human progenitors into neurons with a hindbrain fate, whereas non-functionalized hyaluronic acid favored a forebrain fate. The authors also established that the stiffness of this hybrid material is relevant to cell fate determination; lower elastic moduli, ca. 300 Pa, led to forebrain development, whereas higher elastic moduli, ca. 1000 Pa, led to hindbrain development[116].

Neural organoids have also been cultured in Matrigel-free conditions using microfluidic approaches such as microwell printing. A number of groups were able to generate spheroids in chip-based devices[171–173], but the spheroids lack spatial complexity and cell types compared to fully-developed organoids. Without having to functionalize the well substrates, Chen and coworkers used a 3D printed mold to cast polydimethylsiloxane (PDMS) microwells to generate human embryoid bodies that have the potential to differentiate into brain organoids in a suspension culture[174]. They found that a critical factor affecting differentiation was the ridges of the culture vessel, unlike previous studies using smooth wells. This study represents a new direction towards the generation of organoids, where the shape of the culture vessel might be tuned in order to change cellular phenotype, growth, and differentiation.

Pancreatic organoids have proven difficult to prepare without resorting to Matrigel-based culture. To our knowledge, Candiello and colleagues were the first group to use a synthetic hydrogel, amikacin hydrate crosslinked with poly(ethylene glycol) diglycidyl ether known as Amikagel, to culture hESC-derived islet organoids[175]. Amikagels with elastic moduli ranging from 37 to 320 kPa were created, but no chemical signaling peptides were incorporated into the gels. The authors found that stiffer gels drove pancreatic progenitor cells to aggregate, leading to increased differentiation and maturation into beta-like cells; this may have been mediated through paracrine signaling enhanced by cellular proximity. Compared to cells grown in Matrigel, beta-like cells grown in Amikagel produced higher levels of functional beta-cell markers PDX1 and NKX6.1 and were more responsive to a glucose challenge. Their finding challenges the conventional wisdom that cell-binding domains are required for effective organoid formation. The stiffness of the gel is also very high, exceeding the elastic modulus of materials typically used in the culture of bone[51]. Mechanistically, the authors propose that the culture system is forcing the formation of a compressed organ, rather than serving as a mimetic. A potential disadvantage of this approach is limited control of cellular aggregate sizes and attendant consequences of cell viability of no more than five days. The smallest cell aggregates reported by the authors

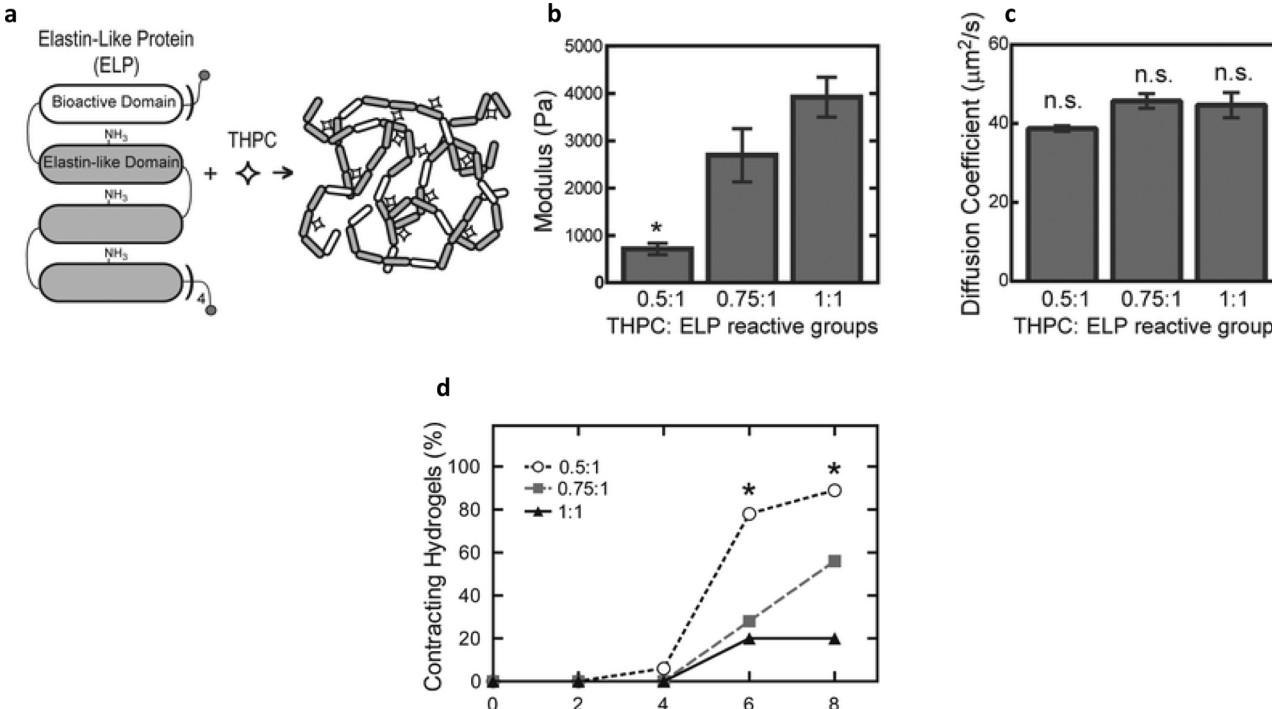

**Fig. 5 Growth of cardiomyocytes on recombinant proteins, and effects of elastic modulus on cardiomyocyte differentiation.** The elastin-like proteins (ELPs) used by Chung and colleagues (**a**) consist of a bioactive domain translationally fused to one or more elastin-like domains; these domains contain lysine groups to facilitate crosslinking by tetrakis hydroxymethyl phosphonium chloride (THPC). By varying the ratio of THPC to ELP reactive groups, it is possible to tune the elastic modulus of the resulting culture matrix (**b**) without significantly altering the diffusion of nutrients or other vital factors through the gel (**c**). Embryoid bodies embedded in the matrix undergo differentiation into cardiomyocytes most favorably in the gels with the lowest elastic modulus (**d**); the cells show the greatest contractility when grown in protein crosslinked with a 0.5:1 ratio of THPC:ELP reactive groups. The image is from ref. [190] and is reproduced with permission.

were ca. 200 microns in diameter; the typical human islet has a diameter of ca. 130 microns[176]. Larger islets have been known to form necrotic centers because of a lack of oxygen and nutrient diffusion; other studies have found that islets best maintain cellular identity and function when they have a size of ca. 100–150 microns[177]. Further work should be done in this system to establish long-term viability.

**Advantages and disadvantages of synthetic polymeric matrices.** A major advantage of using synthetic polymers for organoid culture is that they are amenable to systematic variation in structure and properties and can be used to explore the effects of mechanical and chemical cues on cellular fate[178,179]. Moreover, many such materials, including PEG and PLGA, have been approved by the FDA for use in human therapeutics. Nguyen and colleagues recently assessed more than 1200 synthetic polymer formulations for toxicity and abilities to promote implant vascularization and endothelial cell network formation[180]. This work provides a valuable resource for organoid researchers, particularly those concerned with vascularization of organoids postimplantation.

There are several disadvantages of synthetic hydrogels. First, many synthetic hydrogels require the incorporation of biochemical cues such as cell-binding peptides. In the absence of biochemical cues cells may not attach to the hydrogel, leading to anoikis (a type of programmed cell death)[181] instead of organoid formation[162]. Improper spacing of biochemical cues can also lead to cell death[182]. While the backbone materials of synthetic hydrogels are cheap and can be produced on an industrial scale, functionalization of these materials with precisely-placed, custom-made peptides significantly increases

cost and requires expertise in materials science, making these materials less attractive to cell biology labs. There have been a number of studies showing that organoids can be grown on unmodified surfaces such as alginate, but this requires more research[103,105,110,175,183]. Further, synthetic hydrogels may degrade into cytotoxic by-products[184] or require cytotoxic initiators[185], limiting the types of polymers that can be used in cell culture[186]. Synthetic hydrogels may contain pendant or other unreacted groups, which may be toxic to cells (such as neurotoxic maleimides)[187]. Finally, synthetic hydrogels used as medical implants can trigger foreign body reactions[188]; similar effects may be seen in immunogen-containing organoids. For these reasons, it may be advantageous to engineer recombinant protein gels.

**Organoid culture in peptide and recombinant protein matrices**
Recombinant proteins made by genetically engineered organisms have found wide applications in medicine, food processing, and catalysis. Engineered recombinant protein gels possess major advantages compared to other culture methods: chemical cues can be added with exact definition; chemical and mechanical properties of the gel can be altered independently; polydispersity is low; and degradation rates can be programmed by including appropriate recognition sites for MMP degradative enzymes.

Chung and colleagues generated a hydrogel using an elastin-like polypeptide matrix containing a fibronectin-derived RGD cell-binding domain with tetrakis(hydroxymethyl)phosphonium chloride (THPC) as an amine-reactive crosslinker; this material transiently inhibited contractility of murine ESC-derived

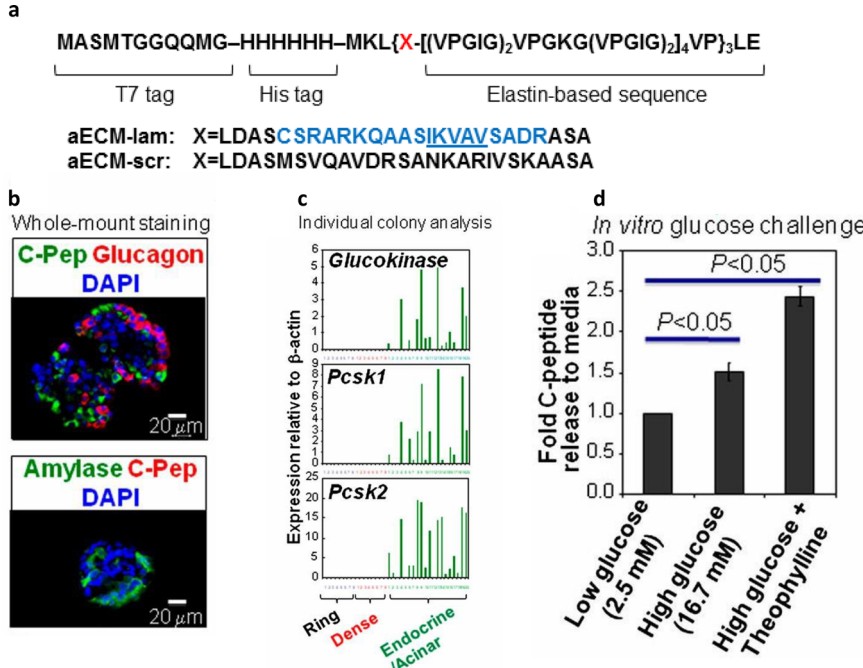

**Fig. 6 Generating pancreatic organoids with a recombinant ECM protein.** An artificial elastin-like polypeptide functionalized with a sequence from laminin can be used to generate organoids from pancreatic ductal progenitor cells from adult mice. (**a**) The recombinant protein (named aECM-lam) incorporates an IKVAV-containing 18-amino acid sequence derived from α1 laminin. The aECM-scr is a scrambled sequence control for aECM-lam. (**b**) aECM-lam permits the differentiation of endocrine (expressing C-peptide and glucagon) and acinar cell lineages (expressing amylase). (**c**) Individual organoids (Endocrine/Acinar) grown in aECM-lam express beta-cell maturation markers glucokinase, *Pcsk1*, and *Pcsk2*. (**d**) Organoids grown in aECM-lam are capable of secreting insulin in vitro when challenged by high concentrations of D-glucose or a combination of D-glucose and cAMP activator theophylline. The image is from ref. [192] and is reproduced with permission.

cardiomyocytes and enhanced survival of dorsal root ganglia cells from chick embryos[189]. In a follow-up study, murine cardiomyocyte differentiation (measured by α-myosin expression, cell contractility, and metabolic activity) was found to be dependent on the stoichiometric ratio of THPC to protein, which tuned the stiffness of the hydrogel. Among the elastic moduli studied (700, 3000, and 4000 Pa), the softest material favored the proliferation of embryoid bodies that contain mesodermal progenitor cells and promoted rapid cardiomyocyte differentiation (Fig. 5). Embryoid bodies cultured in the 700 Pa matrix displayed the highest level of MMP secretion. Inhibition of MMP secretion was deleterious to proliferation and differentiation, suggesting that remodeling of the matrix is essential in cardiomyocyte differentiation[190].

In another follow-up study, a very soft matrix with an elastic modulus of 180 Pa promoted intestinal organoid-forming efficiencies comparable to those observed in collagen I-based matrices. Organoid-forming efficiency was higher when the engineered ECM proteins contained 3.2 mM RGD peptide, compared to no RGD. Interestingly, MMP activity was significantly higher in the stiffer matrices. Inhibition of MMP activity reduced organoid-forming efficiency in the stiffer engineered hydrogel matrices, suggesting that secretion of degradative enzymes in adult intestinal organoids may be a response to overly stiff conditions[191].

Recombinant ECM protein has been investigated for pancreatic organoid culture. The Tirrell and Ku groups have jointly developed and studied an artificial elastin-like polypeptide that incorporates an 18-amino acid sequence derived from α1 laminin; this polypeptide has been named artificial (a) ECM-lam (Fig. 6). aECM-lam was used to supplement a methylcellulose-based 3D pancreatic organoid culture that was devoid of Matrigel. Adult murine Sox9/EGFP+ ductal progenitor cells were first proliferated in Matrigel, then transferred to a culture containing aECM-lam but not Matrigel. After 2 weeks, endocrine-acinar organoids were observed, demonstrating that aECM-lam was capable of inducing differentiation of ductal progenitor cells into endocrine and acinar lineages[192]. A follow-up study established that when Matrigel was added, endocrine and acinar cell development was inhibited while ductal cell formation was promoted[193], demonstrating the importance of the ECM microenvironment in pancreatic organoid differentiation. Using aECM-lam, other morphologically-distinct organoids were formed from murine postnatal pancreas[194] and sorted adult ductal progenitor cells[195]. Finally, the exact population of adult progenitor cells capable of giving rise to endocrine/acinar cells in aECM-lam was determined to be ductal cells, which have high levels of CD133 but low levels of CD71 expression[195]. Collectively, these studies demonstrate the utility of aECM-lam in promoting endocrine and acinar cell differentiation in pancreatic organoid culture and identifying the responsible progenitor population.

Peptide-based hydrogels have recently been employed to model Alzheimer's disease. Zhang and colleagues used the self-assembling peptide RADA-16 to culture human neuronal cells treated with exogenous amyloid-β oligomers, known contributors to Alzheimer's disease. A 3D culture in RADA-16 resulted in activation of a p21-activated kinase in response to amyloid-β oligomers. Both the activation and localization patterns of the p21-activated kinase are characteristic of neurons in an Alzheimer's disease state. In contrast, the corresponding 2D culture did not show this activation and localization, suggesting that the 3D organoid culture of neurons is critical for modeling Alzheimer's disease[196].

The HYDROSAP self-assembling peptide hydrogel is a system recently developed by Pugliese, Marchini, and colleagues. In this system, multi-functionalized and branched self-assembling peptides (SAPs) can generate hydrogels with controllable

elastic moduli[197]. The authors used HYDROSAP peptide 3D hydrogels with elastic moduli of ~800 Pa (similar to the stiffness of human brain tissue) to culture human fetal neural stem cells[198], which were able to differentiate into various lineages including astrocytes, oligodendrocytes, and neurons.

Related work has been performed by Edelbrock and colleagues, with peptide amphiphiles capable of forming long, self-assembled nanostructures within a hydrogel. The peptides contain brain-derived neurotrophic factor (BDNF), which enables the formation of mature neurons via activation of the TrkB pathway. Display of BDNF on the peptide amphiphile is necessary for this effect to be observed[199]. Similar work has shown promise in stem cell differentiation and neural regeneration following spinal injury in vivo[200–202], further demonstrating the utility of peptide-based materials in cell culture.

**Advantages and disadvantages of recombinant protein matrices**. Self-assembling peptides and recombinant proteins offer important advantages in organoid culture. Recombinant proteins are molecularly well-defined and can be tuned independently for stiffness, viscoelastic behavior, and chemical functionality[203–205]. They can be programmed to degrade and remodel at controlled rates by including protease recognition sites[206] or changing crosslinking chemistry[207]. Protein-based hydrogels can be outfitted with a broad range of chemical functionalities by introducing noncanonical amino acids[208,209]; they can also be readily tailored to a wide variety of biomedical contexts[210,211] and made thermally responsive[212,213]. The programmability of recombinant proteins has prompted increasing interest in the design of protein-based hydrogels as matrices for organoid culture.

Protein-based materials have several disadvantages. First, not all proteins can be recombinantly expressed and ensuring re-folding and functionality of these proteins can be challenging. Certain recombinant proteins and self-assembling peptides are immunogenic[214–218]. Ensuring that the recombinant protein is of human origin does not guarantee non-immunogenicity[219]. Care must be taken to avoid introducing other immunogenic factors, such as bacterial endotoxin. Therefore, proteins for clinical use would preferably be expressed in mammalian expression systems (e.g., Chinese Hamster Ovary) or in yeasts (e.g., Pichia pastoris).

**Outlook and conclusions**
Although several Matrigel-free techniques have been developed, they have been used in a narrow range of target tissues; expanding the number of tissue types will increase the acceptance of these alternative techniques. The ideal material for organoid culture should allow independent changes in the chemical and mechanical properties so that the effects on organoid growth, development, or morphology can be correlated. It should also functionalize biologically-relevant cell-binding proteins or peptides with ease. Finally, the ideal material should mimic the dynamic nature of the ECM in terms of erosion rate, viscoelasticity, and susceptibility to degradation. Because of these requirements, synthetic materials and programmable recombinant proteins represent fruitful areas of future research.

This review has proceeded with the assumption that a matrix is required to culture organoids, but matrix-free culture systems have also been developed. For instance, Pagliuca and colleagues kept human embryonic stem cells suspended in liquid culture at 70 rpm, added specific growth factors to encourage differentiation, and found that the resulting beta-like cells behaved similarly to mature beta cells[220]. A similar process was used by Nair and colleagues using mechanical agitation to keep human cells suspended in culture, resulting in functional beta-like clusters[221]. Control over the mechanical environment in such a system could be exerted by changing the speed of agitation. This method has the advantage of allowing easy harvesting of cells, which are simply allowed to settle to the bottom of a tube.

Using the methods discussed above, we anticipate a gradual shift away from the use of Matrigel in organoid culture and towards methods that enable exact control of the cell's mechanical and chemical environments with a more precise definition.

**Reporting Summary**. Further information on research design is available in the Nature Research Reporting Summary linked to this article.

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

## Acknowledgements

We thank Elena C. Chen for assisting in graphic illustration. We also thank Prof. David A. Tirrell for helpful discussions and editing assistance. M.T.K. was supported by the Department of Defense through the National Defense Science & Engineering Graduate (NDSEG) Fellowship Program, and H.T.K. was supported by National Institutes of Health Grant R01DK099734. Support from The Wanek Family Project for Type 1 Diabetes to H.T.K. is also gratefully acknowledged. The content is solely the responsibility of the authors and does not necessarily represent the official views of the National Institutes of Health.

## Author contributions

Conception: M.T.K. and H.T.K.; Writing: M.T.K., H.T.K., and C.J.C.

## Competing interests

The authors declare no competing interests.
