## [Peer Review File · Communications Biology]

Reviewers' comments:

Reviewer #1 (Remarks to the Author):

Matrigel has been a traditional material for culturing organoids, but bears many disadvantages. This manuscript introduced a number of alternative methods to create organoids, in avoidance of the use of Matrigel. The topic is very interesting and timely. However, there are a few issues that need to be addressed before being accepted.

Major concerns

1. This review covered the fabrication of various organoids, such as the liver, gut, brain, and pancreas, but only listed the examples for each without introducing these fundamentally different organoids. This gave the reader a false impression that all organoids are similar. Indeed, they are quite different in both cell compositions and tissue structures. It would be important to have an additional section to detail all organoid types and point out the key features. Accordingly, an additional figure would be also desired to illustrate these organoids' structures.
2. Overall, this manuscript falls short in illustrations, especially those that can demonstrate the results of the cited studies. It is strongly recommended to add 2-3 figures for this purpose.
3. The topic is Matrigel-free, but in general this manuscript compares the alternative methods. It is desired to compare the alternative methods to the traditional Matrigel method.

Minor concerns

4. Figure 1 is interesting, but lacks detailed information, such as the identity of the signal molecules and how they are transduced into cells and affect organoid-related pathways. It would be desired to implement these details in the figure. There are the same issues with Figure 2.
5. In "Synthetic hydrogels" section, chemical and mechanical cues of the synthetic materials are widely discussed in this topic. However, the mechanical cues are only discussed regarding the stiffness. However, synthetic materials' porosity affects the stress relaxation and stress stiffening effects on organoid differentiation, which are not discussed. Hence, discussion on how the porosity affects the organoid differentiation would be helpful.
6. Are these synthetic materials' elasticity similar to Matrigel and how degradable they would be beneficial information? Also, direct comparison of elastic and viscoelastic synthetic materials that can be used for different organoid cultures, and their effect on organoid differentiation should be discussed further.
7. It is worth noticing that physical methods have also emerged to avoid Matrigel, such as one shown in a recently published article "A Matrigel-free method to generate matured human organoids".
8. The authors talk a lot about mechanical stiffness and its effect. It would be desired to add a table clearly stating which mechanical stiffness is suitable for which organoid culture (such as liver, brain, intestine, and pancreas, etc.), and which hydrogel is recommended based on their literature knowledge.
9. Section 3.1 Second paragraph, first line, "Cell binding....for organoid growth and differentiation," needs references.
10. Section 3.3 Modeling the brain and the blood-brain barrier - this topic seems to be distanced from other subsections. It is recommended to single it out and make a separate section.
11. Section 4.1 Paragraph one, line one, Need a reference.
12. Here, every topic from decellularized matrices to synthetic to peptide matrices were discussed as one is better over the other. Along the way, I feel the direct comparison to Matrigel is missing. Also, why Matrigel is still used for organoid culture and not completely avoided. Therefore, "Without Matrigel" the title seems to be overstating.
13. cerebral organoids using 3D-printed microwell arrays". It is recommended to discuss these studies and cite related articles.

Reviewer #2 (Remarks to the Author):

Overall, the review is too long. It includes too many studies that are not relevant to the topic and the authors give very superficial descriptions of the studies referenced. In my opinion, the review would be much better and easier to read if it focused just on organoids (which would decrease the

number of references) and if the descriptions of the main findings of papers cited were more extensive and more critically reviewed. I would reconsider the review suitable for publication after some revisions.

Suggestions

I would suggest changing "questions in basic and developmental biology, disorders," "to questions in basic and developmental biology, genetic disorders,".

Change "we discuss Matrigel-free methods for the creation of organoids" to "we discuss Matrigel-free methods for the generation and maintenance of organoids".

patient-specific adult and patient-derived tissues is redundant.

"In an early report of organoid culture, Clevers and colleagues grew human intestinal Lgr5+ cells in high concentrations of Matrigel supplemented with the growth factors WNT, Noggin, R-spondin, and EGF.¹⁴" This work was done by Sato and colleagues and was one with mouse (not human) Lgr5-eGFP+ stem cells.

The sentence which begins with "For example, gut organoids cultured in Matrigel" is inaccurate. Villi formation requires mesenchymal cells which most organoid systems lack. There is no evidence that Laminin 511 alone can induce villus morphogenesis.

Immunogenicity should also be mentioned as a limitation of Matrigel.

"If xenogeneic materials are properly prepared, they do not cause a serious immune response;⁴⁸".

From the abstract of that reference "Xenogeneically implanted mice showed an acute inflammatory response followed by chronic inflammation and ultimately graft necrosis, consistent with rejection." Figure 1 should explain what each of the cells in the image are.

The authors which from human intestinal organoids to enteroids without explaining what these terms signify.

The authors also fail to mention that the Ootani method preserves the mesenchymal niche in the organoids.

The authors also cite several examples of 2D culture of cells on matrices. Discussing these studies seems irrelevant to a review on "organoids without Matrigel".

In discussing human intestinal organoids, the authors need to specify if they are biopsy derived or human pluripotent stem cell derived.

The section on blood brain barrier describes a transwell system which is a monolayer system and not really relevant to the topic of the review.

The cell source of pancreatic organoids needs to be stated (pluripotent stem cells or patient biopsy).

"Excitingly, when the cultured cells were injected into a rat model of spinal cord injury, the rats recovered from the injury." Define recovered? How was recovery measured?

Reviewer #3 (Remarks to the Author):

In "Organoids without Matrigel," Kozlowski, Crook, and Ku review current state of the art in culturing and manipulating organoids without the use of EHS-tumor derived matrix, the most popular of which is Matrigel by Corning. While Matrigel has been a valuable tool for researchers, its variability, undefined nature, and xenogeneic sourcing preclude it from being incorporated in many applications. Covered in this review are organoid-based applications of different sources of matrix ranging from natural sources (decellularized ECM, purified ECM proteins, polysaccharides), synthetic sources (e.g. PEG-based), and recombinant peptide-based systems. The article is well-written with discussion of both immediately relevant literature and future applications such as transplantation and complex body-on-a-chip technologies. Below are minor content suggestions and critiques.

Comments

1. Authors should double check that all references are in the appropriate place. For example, in Table 1, Reference 64 is indicated to refer to testicular ECM and organoid culture, however the correct reference is 65. References 65 and 66 are listed for pancreas, the correct references are 66 and 67.

2. A reference authors could consider including in Section 2.1. This article uses a LEMgel to grow

branching bile duct networks, and compares different cell and matrix types for their ability to induce branch formation.

a. Lewis et al. "Complex bile duct network formation within liver decellularized extracellular matrix hydrogels." *Scientific Reports*. 2018.

3. An additional reference for Section 2.2. This paper included proteomics analysis of ECM, as well as multiple endodermal organoid (liver, intestine, bile duct, stomach) culture.

a. Giobbe et al. "Extracellular matrix hydrogel derived from decellularized tissues enables endodermal organoid culture." *Nature Communications*. 2019.

4. An additional reference for Section 2.3. This paper includes proteomics comparison of ECM and Matrigel for the purposes of pancreatic islet and beta cell differentiation.

a. Bi et al. "Proteomic analysis of decellularized pancreatic matrix identifies collagen V as a critical regulator for islet organogenesis from human pluripotent stem cells." *Biomaterials*. 2019.

5. In section 3.5. "Advantages and disadvantages of synthetic polymeric matrices," authors discuss several disadvantages of alginate. Alginate is a naturally-derived polymer, being a polysaccharide derived from brown algae. Although it is subject to the limitations mentioned, these discussions may be more appropriate for section 2.5. The paper by Capeling et al may also be suited for additional discussion in 3.2. as different alginate weight percentages (and therefore mechanical properties) imparted different effect on HIOs.

6. Section 3.5. should also mention that modifying synthetic hydrogels with peptide epitopes (e.g. for cell binding or MMP degradation) can significantly increase cost, preventing new labs from adopting the technology. Cost is of course at a tradeoff of precisely defined chemical structures.

7. Authors should consider adding this additional reference for Section 4.3. or 4.4. This paper describes a self-assembled peptide amphiphile supramolecular matrix that enables brain-derived neurotrophic factor (BDNF) bioactivity, compared to soluble BDNF, and supports cortical neuron infiltration.

a. Edelbrock et al. "Supramolecular Nanostructure Activates TrkB Receptor Signaling of Neuronal Cells by Mimicking Brain-Derived Neurotrophic Factor." *Nano Letters*. 2018.

8. Achieving 3D organization in some organoid platforms does not always require the use of Matrigel or a gel at all. Many embryoid body-based differentiation protocols, although dated and inefficient, did not use Matrigel embedding. Several approaches use culture at an air-liquid interface or suspension culture in spinner flasks. This could be mentioned briefly in the introduction or conclusion sections. The authors should consider including recent stem cell and organoid papers that do not use an embedding matrix:

a. Takasato et al. "Kidney organoids from human iPS cells contain multiple lineages and model human nephrogenesis." *Nature Letters*. 2015.

b. Pagliuca et al. "Generation of Functional Human Pancreatic β Cells In Vitro." *Cell*. 2014.

c. Nair et al. "Recapitulating endocrine cell clustering in culture promotes maturation of human stem-cell-derived β cells." *Nature Cell Biology*. 2019.

Minor comments

9. Figures 3 and 4 are overlapped due to formatting error in the PDF file.

Response to Reviewer #1:

Matrigel has been a traditional material for culturing organoids, but bears many disadvantages. This manuscript introduced a number of alternative methods to create organoids, in avoidance of the use of Matrigel. The topic is very interesting and timely. However, there are a few issues that need to be addressed before being accepted.

Major concerns

1. This review covered the fabrication of various organoids, such as the liver, gut, brain, and pancreas, but only listed the examples for each without introducing these fundamentally different organoids. This gave the reader a false impression that all organoids are similar. Indeed, they are quite different in both cell compositions and tissue structures. It would be important to have an additional section to detail all organoid types and point out the key features. Accordingly, an additional figure would be also desired to illustrate these organoids' structures.

We thank the reviewer for bringing up this important point. Due to word limit, we have added references to direct the readers to more detailed reviews on each of the organoid types discussed in this manuscript.

2. Overall, this manuscript falls short in illustrations, especially those that can demonstrate the results of the cited studies. It is strongly recommended to add 2-3 figures for this purpose.

We thank the reviewer for this concern and have added 2 figures: one figure detailing the use of liver decellularization, and one figure detailing the use of synthetic hydrogel matrices, and the effects of stiffness on cellular differentiation.

3. The topic is Matrigel-free, but in general this manuscript compares the alternative methods. It is desired to compare the alternative methods to the traditional Matrigel method.

We are sorry that we did not make this point clear in the original manuscript. We now have explicitly stated and compared Matrigel with alternative methods whenever the referenced studies relate them.

Minor concerns

4. Figure 1 is interesting, but lack of detailed information, such as the identity of the signal molecules and how they are transduced into cells and affect organoid-related pathways. It would be desired to implement these details in the figure. There are same issues with Figure 2.

We have updated Figure 1 to include more details as requested. The original figure 2 on blood brain barrier has been eliminated due to word limit.

5. In "Synthetic hydrogels" section, chemical and mechanical cues of the synthetic materials are widely discussed in this topic. However, the mechanical cues are only discussed regarding the stiffness. However, synthetic materials' porosity affects the stress relaxation and stress stiffening effects on organoid differentiation, which are not discussed. Hence, discussion on how the porosity affects the organoid differentiation would be helpful.

As suggested, we have added the information on porosity in this manuscript.

6. Are these synthetic materials' elasticity similar to Matrigel and how degradable they would be beneficial information? Also, direct comparison of elastic and viscoelastic synthetic materials that can be used for different organoid cultures, and their effect on organoid differentiation should be discussed further.

As suggested, we have added the information on elasticity and degradability of synthetic materials in this manuscript.

7. It is worth of noticing that physical method has also emerged to avoid Matrigel, such as one shown in a recently published article “A Matrigel-free method to generate matured human cerebral organoids using 3D-printed microwell arrays”. It is recommended to discuss these studies and cite related articles.

As suggested, we have added this article for discussion in section 3.3.

8. The authors talk a lot about mechanical stiffness and its effect. It would be desired to add a table clearly stating which mechanical stiffness is suitable for which organoid culture (such as liver, brain, intestine, and pancreas, etc.), and which hydrogel is recommended based on their literature knowledge.

As suggested, we have added a new table 2 to list studies on stiffness and organoid culture. The studies listed in this table are also further discussed in the text.

9. Section 3.1 Second paragraph, first line, "Cell binding...for organoid growth and differentiation," needs references.

We are sorry that we did not make our writing clear in the original manuscript. We have re-written this sentence to make it clear that we are referring to the references that follow the first sentence.

10. Section 3.3 Modeling the brain and the blood-brain barrier - this topic seems to be distanced from other subsections. It is recommended to single it out and make a separate section.

We have deleted this section in response to the editor’s request to focus on organoids and reduce the length of the manuscript.

11. Section 4.1 Paragraph one, line one, Need a reference.

We are sorry that we did not make our writing clear in the original manuscript. We have re-written this sentence as follows: “As many studies, such as those cited in section 3.2, have emphasized, the mechanical properties of ECM can have significant effects on organoid growth and development.”

12. Here, every topic from decellularized matrices to synthetic to peptide matrices were discussed as one is better over the other. Along the way, I feel the direct comparison to Matrigel is missing. Also, why Matrigel is still used for organoid culture and not completely avoided. Therefore, "Without Matrigel" the title seems to be overstating.

We acknowledge the reviewer’s point that Matrigel is still being used heavily in the field of organoid culture and that Matrigel is difficult to replace at the current time. In this context, the studies we reviewed in this article are the beginning attempts to replace Matrigel using various materials. Therefore, we agree with the reviewer’s assessment and have changed the title to “Towards organoid culture without Matrigel”. The summary for the comparison of Matrigel to other materials is now included in Table 1.

Response to Reviewer #2:

Overall, the review is too long. It includes too many studies that are not relevant to the topic and the authors give very superficial description of the studies referenced. In my opinion, the review would be much better and easier to read if it focused just on organoids (which would decrease the number of references) and if the descriptions of the main findings of papers cited were more extensive and more critically reviewed. I would reconsider the review suitable for publication after some revisions.

We agree. We have now focused this review to organoid and have cut down the section on blood-brain barrier and other studies that do not directly relate to 3D organoid culture.

Suggestions

1. I would suggest changing “questions in basic and developmental biology, disorders,” “to questions in basic and developmental biology, genetic disorders,”.

Thank you for this suggestion. We have updated the language in the abstract accordingly.

2. Change “we discuss Matrigel-free methods for the creation of organoids” to “we discuss Matrigel-free methods for the generation and maintenance of organoids”.

We have updated this language in the abstract.

3. patient-specific adult and patient-derived tissues is redundant.

We have updated this language in the introduction section.

4. “In an early report of organoid culture, Clevers and colleagues grew human intestinal Lgr5+ cells in high concentrations of Matrigel supplemented with the growth factors WNT, Noggin, R-spondin, and EGF.14” This work was done by Sato and colleagues and was one with mouse (not human) Lgr5-eGFP+ stem cells.

We have updated the citation to read as Sato et al.

5. The sentence which begins with “For example, gut organoids cultured in Matrigel” is inaccurate. Villi formation requires mesenchymal cells which most organoid systems lack. There is no evidence that Laminin 511 alone can induce villus morphogenesis.

We are sorry that we did not make the writing clear. As suggested, we now have changed the wording accordingly to “For example, gut organoids cultured in Matrigel lack the characteristic architecture of mammalian intestines, which could be due to a sub-optimal amount of laminin-511 and the absence of other cell types such as mesenchymal cells.”

6. Immunogenicity should also be mentioned as a limitation of Matrigel.

We have added this detail in Introduction section, as well as addressed in Table 1.

7. “If xenogeneic materials are properly prepared, they do not cause a serious immune response;48”. From the abstract of that reference “Xenogeneically implanted mice showed an acute inflammatory response followed by chronic inflammation and ultimately graft necrosis, consistent with rejection.”

We apologize that we did not make clear the message of this referenced study by Allman et al. We have clarified it as follows. “While xenogeneic materials have the potential to cause immune responses, this risk can be greatly reduced or eliminated by using proper preparation techniques; similar decellularized ECM scaffolds are FDA-approved for many clinical applications such as heart valve replacement, facial reconstruction, and osteopathic implants.”

8. Figure 1 should explain what each of the cells in the image are.

We have updated Figure 1 to include additional details.

9. The author switch from human intestinal organoids to enteroids without explaining what these terms signify.

Thank you for pointing this out. We have added definitions in the text as follows. “HIOs are organoids derived from pluripotent stem cells (PSCs), and enteroids are derived from adult crypt stem cells”

10. The authors also fail to mention that the Ootani method preserves the mesenchymal niche in the organoids.

We have added this information in the text as follows. “In contrast to the TMDU method, the Ootani method uses a collagen I gel in which small and large intestinal cells are kept suspended at an air-liquid interface; this method improves oxygenation of the organoid, allows viable murine organoids to be maintained in culture for up to 350 days, and preserves the mesenchymal niche in the organoids”.

11. The authors also cite several examples of 2D culture of cells on matrices. Discussing these studies seems irrelevant to a review on “organoids without Matrigel”.

We have removed the references to these 2D studies to strengthen the focus of our manuscript on 3D organoid culture.

12. In discussing human intestinal organoids, the authors need to specify if they are biopsy derived or human pluripotent stem cell derived.

This point is resolved after we clearly define the terms “human intestinal organoids (HIO)” vs. “enteroids”. Please see response to #9 above.

13. The section on blood brain barrier describes a transwell system which is a monolayer system and not really relevant to the topic of the review.

We agree. We have eliminated this section.

14. The cell source of pancreatic organoids needs to be stated (pluripotent stem cells or patient biopsy).

We have added this information in the text.

15. “Excitingly, when the cultured cells were injected into a rat model of spinal cord injury, the rats recovered from the injury.” Define recovered? How was recovery measured?

Thank you for pointing this out. We have clarified the language of these results in the text as follows.

“When differentiated organoids were injected together with HYDROSAP peptides into a rat model of spinal cord injury, the rats recovered from the injury better than a comparative sham treatment. Rats were injured by a surgical incision

between the T9 and T10 thoracic vertebrae. Recovery from the spinal cord injury was scored according to the Basso, Beattie, and Bresnahan (BBB) locomotor rating scale, which is used to rate locomotion based on factors such as paw movement on a scale of 0-21.”

We also have added a few sentences to caution the interpretation of the data of this study.

Response to Reviewer #3:

In “Organoids without Matrigel,” Kozlowski, Crook, and Ku review current state of the art in culturing and manipulating organoids without the use of EHS-tumor derived matrix, the most popular of which is Matrigel by Corning. While Matrigel has been a valuable tool for researchers, its variability, undefined nature, and xenogeneic sourcing preclude it from being incorporated in many applications. Covered in this review are organoid-based applications of different sources of matrix ranging from natural sources (decellularized ECM, purified ECM proteins, polysaccharides), synthetic sources (e.g. PEG-based), and recombinant peptide-based systems. The article is well-written with discussion of both immediately relevant literature and future applications such as transplantation and complex body-on-a-chip technologies. Below are minor content suggestions and critiques.

Comments

1. Authors should double check that all references are in the appropriate place. For example, in Table 1, Reference 64 is indicated to refer to testicular ECM and organoid culture, however the correct reference is 65. References 65 and 66 are listed for pancreas, the correct references are 66 and 67.

Thank you for pointing this out. We have now carefully matched the appropriate references and updated the citations as needed.

*2. A reference authors could consider including in Section 2.1. This article uses a LEMgel to grow branching bile duct networks, and compares different cell and matrix types for their ability to induce branch formation.
a. Lewis et al. “Complex bile duct network formation within liver decellularized extracellular matrix hydrogels.” Scientific Reports. 2018.*

Thank you for suggesting this paper and those below. We have added this study in Section 2.1.

*3. An additional reference for Section 2.2. This paper included proteomics analysis of ECM, as well as multiple endodermal organoid (liver, intestine, bile duct, stomach) culture.
a. Giobbe et al. “Extracellular matrix hydrogel derived from decellularized tissues enables endodermal organoid culture.” Nature Communications. 2019.*

We have added this citation in Section 2.2.

*4. An additional reference for Section 2.3. This paper includes proteomics comparison of ECM and Matrigel for the purposes of pancreatic islet and beta cell differentiation.
a. Bi et al. “Proteomic analysis of decellularized pancreatic matrix identifies collagen V as a critical regulator for islet organogenesis from human pluripotent stem cells.” Biomaterials. 2019.*

We have added this citation in Section 2.3.

5. In section 3.5. “Advantages and disadvantages of synthetic polymeric matrices,” authors discuss several disadvantages of alginate. Alginate is a naturally-derived polymer, being a polysaccharide derived from brown algae. Although it is subject to the limitations mentioned, these discussions may be more appropriate for section 2.5. The paper by Capeling et al may also be suited for additional discussion in 3.2. as different alginate weight percentages (and therefore mechanical properties) imparted different effect on HIOs.

As suggested, we have moved the majority of the discussion on alginate to Section 2.4. (Please note that due to re-focusing of this manuscript, numbers for certain sections have changed.) We have also briefly discussed Capeling in Section 3.4 as relates to the possibility of using non-functionalized materials as platforms for growth.

6. Section 3.5. should also mention that modifying synthetic hydrogels with peptide epitopes (e.g. for cell binding or MMP degradation) can significantly increase cost, preventing new labs from adopting the technology. Cost is of course at a tradeoff of precisely defined chemical structures.

We have discussed this information in Section 3.4.

7. Authors should consider adding this additional reference for Section 4.3. or 4.4. This paper describes a self-assembled peptide amphiphile supramolecular matrix that enables brain-derived neurotrophic factor (BDNF) bioactivity, compared to soluble BDNF, and supports cortical neuron infiltration.

a. Edelbrock et al. “Supramolecular Nanostructure Activates TrkB Receptor Signaling of Neuronal Cells by Mimicking Brain-Derived Neurotrophic Factor.” *Nano Letters*. 2018.

We have now included this study in Section 4.2.

8. Achieving 3D organization in some organoid platforms does not always require the use of Matrigel or a gel at all. Many embryoid body-based differentiation protocols, although dated and inefficient, did not use Matrigel embedding. Several approaches use culture at an air-liquid interface or suspension culture in spinner flasks. This could be mentioned briefly in the introduction or conclusion sections. The authors should consider including recent stem cell and organoid papers that do not use an embedding matrix:

a. Takasato et al. “Kidney organoids from human iPS cells contain multiple lineages and model human nephrogenesis.” *Nature Letters*. 2015.

b. Pagliuca et al. “Generation of Functional Human Pancreatic β Cells In Vitro.” *Cell*. 2014.

c. Nair et al. “Recapitulating endocrine cell clustering in culture promotes maturation of human stem-cell-derived β cells.” *Nature Cell Biology*. 2019.

This is an excellent point. We have included Pagliuca et al and Nair et al in the conclusion section.

Minor comments

9. Figures 3 and 4 are overlapped due to formatting error in the PDF file.

We are sorry for the oversight. Formatting is now accurate.

Reviewers' comments:

Reviewer #1 (Remarks to the Author):

The authors have well addressed all my previous comments. I recommend for accept the manuscript now.

Reviewer #2 (Remarks to the Author):

In my opinion, the review would be much better and easier to read if the descriptions of the main findings of papers cited were more extensive and more critically reviewed. Also, the manuscript is very repetitive and hard to read. I would reconsider the review suitable for publication after some revisions.

Suggestions

"free of interfering cell types such as epithelial". This statement is wrong since gastrointestinal organoids are epithelial.

In the paragraph starting with "Many organoids have been cultured in Matrigel, a widely-used material derived from the 16 secretion of Engelbreth-Holm-Swarm mouse sarcoma cells", the authors mention the development of protocols for different types of organoids and they do not specify whether they are mouse or human. I would suggest first stating the development of the mouse organoids and then adaptation of these protocols for growth of human organoids.

"In an early report of organoid culture, Sato and colleagues grew human intestinal Lgr5+ cells in high concentrations of Matrigel supplemented with the growth factors WNT, Noggin, R-spondin, and EGF.¹⁴" This work was done with mouse (not human as still stated in manuscript) Lgr5-eGFP+ stem cells.

"Finally, it has become increasingly clear that the mechanical properties of 3D culture media, including some factors that have not been as extensively studied, such as stress relaxation and 1 creep,". Are these really factors from the media or the culture system?

"Matrigel can be prone to rapid precipitation", I think this should be state that it solidifies rapidly not precipitate. Also, this paragraph should just be combined with the previous paragraph.

In table 2 neuron needs to be changed to neuroepithelial. Also reference 158 uses mouse ESCs so I'm not sure why this reference is here.

Also, table 2 needs to specify the species from which the organoids was made.

"Decellularization approaches have been demonstrated for kidney,^{68,69} heart,^{70,71} lung,⁷² and 2 testicular⁷³ tissues", the authors really need to distinguish between mouse and human organoids.

"the resulting tumor organoids 2 had greater hepatocyte function than tumor organoids grown in a comparable collagen I-based culture"

"HIOs are organoids derived from pluripotent stem cells 4 (PSCs), and enteroids are derived from adult crypt stem cells.⁷⁹". This distinction should be made much earlier in the manuscript.

"human pediatric stomach and intestinal cells" I would change to glands since there are very few reports of generating human organoids from single cells.

"Yui and colleagues showed that embedding intact murine intestinal crypts and 33 isolated Lgr5+ progenitor cells in collagen I with hepatocyte growth factor, R-spondin 1, EGF, and Noggin 34 generated organoids that were able to repair damaged mouse intestinal epithelia upon transplantation." These were from colonic crypts and they engrafted and showed some improvement of symptoms.

" β 4 integrins usually arrange themselves on the basal surface of intestinal organoids". Are these integrins expressed from the organoids themselves or are they in the collagen matrix.

"The mechanical properties of alginate can also be easily tuned". What properties can be tuned?

"Finally, the need for pre-differentiation of pluripotent stem cells PSCs or iPSCs into organ-specific progenitor cells means that an additional differentiation step is often necessary for the to grow organoids in decellularized matrix approach to work." What is pre-differentiation? Pluripotent stem cells in a term that encompasses both embryonic stem cells and iPSCs.

"exhibited a higher degree of apical-basal polarity." By what metrics.

Figure 3 panel D has no mention of what the stainings are and what the insets are supposed to show.

"organoid viability at seven days was poor."

"teratoma formation can be suppressed by encapsulating developing neural cells in RGD-functionalized hyaluronan-methylcellulose hydrogels." Are these neural cells in the form of organoids? If not, I don't see how it is relevant to the theme of the manuscript.

There's also too many examples of transplantation and other things completely unrelated to growing organoids without Matrigel.

Reviewer #3 (Remarks to the Author):

The authors have addressed specific comments as well as more general comments towards restructuring and editing the manuscript. Their modifications in response to my and other reviewers' comments, such as shortening certain sections, have improved the overall presentation of the manuscript. The manuscript will be insightful to both the biomaterials and stem cell/organoid communities.

Response to reviewer's comments

Reviewer #1 (Remarks to the Author):

The authors have well addressed all my previous comments. I recommend for accept the manuscript now.

Thank you!

Reviewer #2 (Remarks to the Author):

In my opinion, the review would be much better and easier to read if the descriptions of the main findings of papers cited were more extensive and more critically reviewed. Also, the manuscript is very repetitive and hard to read. I would reconsider the review suitable for publication after some revisions.

Suggestions

1. "free of interfering cell types such as epithelial". This statement is wrong since gastrointestinal organoids are epithelial.

Thank you for pointing this out. We have clarified the sentence to "free of interfering cell types such as vascular, nerve, or undesired epithelial cells".

2. I would suggest first stating the development of the mouse organoids and then adaptation of these protocols for growth of human organoids.

Thank you for this suggestion. We have now specified mouse or human system throughout the manuscript; however, not all human systems originate/adapt from mouse systems.

3. "In an early report of organoid culture, Sato and colleagues grew human intestinal Lgr5+ cells in high concentrations of Matrigel supplemented with the growth factors WNT, Noggin, R-spondin, and EGF.14" This work was done with mouse (not human as still stated in manuscript) Lgr5-eGFP+ stem cells.

Thank you for pointing out this mistake of ours. We have changed the word "human" to "mouse".

4. "Finally, it has become increasingly clear that the mechanical properties of 3D culture media, including some factors that have not been as extensively studied, such as stress relaxation and 1 creep,". Are these really factors from the media or the culture system?

We agree that the word "media" in this context is not as precise as "culture system". We have changed the language wherever is applicable throughout the manuscript.

5. "Matrigel can be prone to rapid precipitation", I think this should be state that it solidifies rapidly not precipitate. Also, this paragraph should just be combined with the previous paragraph.

We agree that "solidification" is a better word than "precipitation" to describe the behavior of Matrigel at 37oC. Due to the request of the editor to cut words, we have removed this sentence because this is a common knowledge among those who use Matrigel in culture and therefore

dispensable.

6. In table 2 neuron needs to be changed to neuroepithelial. Also reference 158 uses mouse ESCs so I'm not sure why this reference is here.

Also, table 2 needs to specify the species from which the organoids was made.

Reference 158 used mouse ESC differentiation towards to MAP2-expressing neuroectodermal cells to discern the needed stiffness in culture system. Reference 159 used mouse ESC derived neuroepithelial cells for study. Accordingly, we have changed "neuron" to "Mouse ESC derived neuroepithelial" in Table 2. We have also added the species info in Table 2.

7. "Decellularization approaches have been demonstrated for kidney,^{68,69} heart,^{70,71} lung,⁷² and 2 testicular⁷³ tissues", the authors really need to distinguish between mouse and human organoids.

Thanks for pointing this out. We have added species info accordingly. The sentence now reads "Decellularization approaches have been demonstrated for human kidney,⁶⁷ murine kidney,^{67,68} murine heart,^{69,70} human and porcine lung,⁷¹ and porcine testicular⁷² tissues, with each type posing unique challenges."

8. "the resulting tumor organoids 2 had greater hepatocyte function than tumor organoids grown in a comparable collagen I-based culture"

We have added the word "tumor" in this sentence, as suggested.

9. "HIOs are organoids derived from pluripotent stem cells 4 (PSCs), and enteroids are derived from adult crypt stem cells.⁷⁹". This distinction should be made much earlier in the manuscript.

We first mentioned the words "intestinal" and "intestines" in the Introduction section, in which Matrigel is being introduced as the major subject of this review. The sentence that reviewer mentioned here is located at section 2.2. entitled "Gut organoids grown in decellularized ECM". In this context, we went further to differentiate the source of cells for intestinal organoids. We believe this is a suitable place to provide such details rather than Introduction.

10. "human pediatric stomach and intestinal cells" I would change to glands since there are very few reports of generating human organoids from single cells.

As suggested, we have changed the word "cells" to "crypts".

11. "Yui and colleagues showed that embedding intact murine intestinal crypts and 33 isolated Lgr5+ progenitor cells in collagen I with hepatocyte growth factor, R-spondin 1, EGF, and Noggin 34 generated organoids that were able to engraft onto damaged mouse intestinal epithelia upon transplantation." These were from colonic crypts and they engrafted and showed some improvement of symptoms.

Thank you for catching this mistake. We have changed "intestinal" to "colonic" crypts.

12. "β4 integrins usually arrange themselves on the basal surface of intestinal organoids". Are these integrins expressed from the organoids themselves or are they in the collagen matrix.

Thank you for pointing this out. We have clarified the sentence as the following, “ β_4 integrin expressed by the intestinal organoids are distributed only at the basal surface while β_1 integrin is known to be required for proper apical-basal polarization.” We have also added a citation to the paper “Controlling Epithelial Polarity: A Human Enteroid Model for Host-Pathogen Interactions.”

13. “The mechanical properties of alginate can also be easily tuned”. What properties can be tuned?

We have clarified the sentence as the following, “The mechanical properties of alginate, such as elastic modulus, extensibility, characteristic relaxation time, can also be easily tuned.”

14. “Finally, the need for pre-differentiation of pluripotent stem cells PSCs or iPSCs into organ-specific progenitor cells means that an additional differentiation step is often necessary for the to grow organoids in decellularized matrix approach to work.” What is pre-differentiation? Pluripotent stem cells in a term that encompasses both embryonic stem cells and iPSCs.

Thank you for pointing this out. We have clarified the sentence as the following, “Finally, the occasional need for PSC differentiation into organ-specific progenitor cells that are then introduced into the decellularized matrix requires an additional step.”

15. “exhibited a higher degree of apical-basal polarity.” By what metrics.

Thank you for pointing this out. We have clarified the sentence as the following, “The optimal conditions (elastic moduli ranged from 2-4 kPa and scaffolds with MMP insensitivity) produce murine neural tube organoids that are more homogenous in colony size and morphology, as well as more polarized than those grown in Matrigel. The percentage of cells containing an actomyosin contractile ring is used as a metric for the polarity of the cells.¹⁵⁷”

16. Figure 3 panel D has no mention of what the stainings are and what the insets are supposed to show.

We have updated the figure legend.

17. “organoid viability at seven days was poor.”

Thank you for pointing this out. We have clarified the sentence as the following, “Intriguingly, when PEG was crosslinked with dithiothreitol to inhibit matrix degradation, organoid viability at seven days was poor as measured by live-dead staining, quantified by the area of the organoid stained as live. This demonstrated a requirement of degradable matrix for prolonged survival.”

18. “teratoma formation can be suppressed by encapsulating developing neural cells in RGD-functionalized hyaluronan-methylcellulose hydrogels.” Are these neural cells in the form of organoids? If not, I don’t see how it is relevant to the theme of the manuscript.

There’s also too many examples of transplantation and other things completely unrelated to growing organoids without Matrigel.

Thank you for reminding us this point, which is in line with the journal's request on reducing total words to 6,000. We therefore have eliminated as much as we can text unrelated to growing organoids without Matrigel.

Reviewer #3 (Remarks to the Author):

The authors have addressed specific comments as well as more general comments towards restructuring and editing the manuscript. Their modifications in response to my and other reviewers' comments, such as shortening certain sections, have improved the overall presentation of the manuscript. The manuscript will be insightful to both the biomaterials and stem cell/organoid communities.

Thank you!